# 'Good health means being mentally, socially, emotionally and physically fit': women's understanding of health and ill health during and after pregnancy in India and Pakistan: a qualitative study

Mary McCauley ![ORCID] ,[1] Ayesha Rasheeda Avais,[2] Ritu Agrawal,[3] Shumaila Saleem,[2] Shamsa Zafar,[4] Nynke van den Broek[1]

¹Centre for Maternal and Newborn Health, Liverpool School of Tropical Medicine, Liverpool, UK
²Child Advocacy International Pakistan, Islamabad, Pakistan
³Vardhman Mahavir Medical College and Safdarjung Hospital, New Delhi, India
⁴Fazaia Medical College, Air University, Islamabad, Pakistan

**Correspondence to**
Dr Mary Mccauley;
mary.mccauley@lstmed.ac.uk

## ABSTRACT

**Objective**  To explore what women consider health and ill health to be, in general, and during and after pregnancy. Women's views on how to approach screening for mental ill health and social morbidities were also explored.

**Settings**  Public hospitals in New Delhi, India and Islamabad, Pakistan.

**Participants**  130 women attending for routine antenatal or postnatal care at the study healthcare facilities.

**Interventions**  Data collection was conducted using focus group discussions and key informant interviews. Transcribed interviews were coded by topic and grouped into categories. Thematic framework analysis identified emerging themes.

**Results**  Women are aware that maternal health is multidimensional and linked to the health of the baby. Concepts of good health included: nutritious diet, ideal weight, absence of disease and a supportive family environment. Ill health consisted of physical symptoms and medical disease, stress/tension, domestic violence and alcohol abuse in the family. Reported barriers to routine enquiry regarding mental and social ill health included a small number of women's perceptions that these issues are 'personal', that healthcare providers do not have the time and/or cannot provide further care, even if mental or social ill health is disclosed.

**Conclusions**  Women have a good understanding of the comprehensive nature of health and ill health during and after pregnancy. Women report that enquiry regarding mental and social ill health is not part of routine maternity care, but most welcome such an assessment. Healthcare providers have a duty of care to deliver respectful care that meets the health needs of women in a comprehensive, integrated, holistic manner, including mental and social care. There is a need for further research to understand how to support healthcare providers to screen for all aspects of maternal morbidity (physical, mental and social); and for healthcare providers to be enabled to provide support and evidence-based care and/or referral for women if any ill health is disclosed.

## Strengths and limitations of this study

► We interviewed 130 women attending for routine antenatal or postnatal care in secondary level public hospitals in two different countries, India and Pakistan.

► We conducted both focus group discussions and key informant semistructured interviews to explore a large number of women's own opinions.

► Similar proportions of women were interviewed at four different stages: (1) first half of pregnancy (≤20 weeks), (2) second half of pregnancy (>20 weeks), (3) early postnatal stage (day 1–7) or (4) late postnatal stage (week 2–6) after childbirth.

► We only interviewed women in urban settings and excluded women living in rural communities who may have alternative perspectives or different insights.

## INTRODUCTION

All women have the right to the highest attainable standard of health and wellbeing including physical, mental and social aspects.[1 2] The Sustainable Development Goals (SDG) envision that every woman in every setting has an equal chance to survive and thrive.[3] SDG 3 is to ensure healthy lives and promote well-being for all.[3] The Global Strategy strives for a world in which every mother can enjoy a wanted and healthy pregnancy and childbirth, and that pregnancy is an opportunity for a woman to receive care to thrive.[1] Currently, this is not the case for many women living in low-income and middle-income countries (LMICs). As part of the need to develop innovative strategies to achieve SDG 3, there has been a renewed focus on maternal ill health or maternal morbidity in terms of definition, measurement and interventions; and especially

for women living in LMIC.[4–10] The burden of disease of various aspects of health (physical, mental and social) during and after pregnancy and their impact on women and/or pregnancy outcomes are currently not well documented in a comprehensive, standardised approach.[4 5] It is suggested that although maternal ill health occurs across all countries, cultures, religions, socioeconomic status and ages, the burden of disease is higher in women living in LMIC, with an estimated three out of four women suffering at least one form of maternal morbidity during or after pregnancy.[4 5] However, there is currently lack of literature regarding the understanding of what women consider maternal morbidity or ill health to be during and after pregnancy across many LMIC.

Maternal ill health or morbidity is a complex and broad concept and can be understood and described in various ways depending on different perspectives.[8–10] For example women, men, families, communities, healthcare providers and policy makers may have mixed views and perceptions on what maternal ill health is and the effect or risk of maternal morbidity during and after pregnancy for the women and/or her baby.[9] Maternal morbidity has been defined as 'any health condition that is attributed to or aggravated by pregnancy and childbirth which has a negative impact on the woman's well-being'.[8] However, there is a lack of understanding and lack of data regarding what women themselves consider ill health to be (physical and/or mental and/or social) and their experiences (subjective measures) compared with clinical findings (objective measures) when assessed and defined as morbidity by a healthcare provider; and there is especially a lack of data from women living in LMIC.[4 5 9] The World Health Organization (WHO) has recently highlighted the need for qualitative studies that explore women's lived experiences of ill health and perceptions of their health and health needs during and after pregnancy.[8–10] In many high-income countries, women's ideas, concerns, expectations and choices regarding their care during and after pregnancy are well considered. However, there is lack of data of women's voices and views regarding what they consider health and well-being to be during and after pregnancy in many LMIC. An understanding of women's perceptions and opinions is essential to inform how healthcare providers can better deliver best care possible within existing healthcare systems in LMIC settings.

This study, therefore, sought to explore what women attending for routine antenatal care (ANC) or postnatal care (PNC) at healthcare facilities in India and Pakistan consider health and ill health to be in general, and, what they consider health and ill health to be, during and after pregnancy. As a component of maternal ill health, women's views on how best to approach aspects of mental ill health (such as depression and/or anxiety) and social ill health (such as exposure to violence and/or substance misuse), were assessed.

## MATERIALS
### Study design
Experienced female researchers (ARA, RA and SS) interviewed a total of 130 women. Data collection was conducted using focus group discussions (FGDs) to interview 60 women attending for routine ANC or PNC in the outpatient clinics at a large public hospital in New Delhi, India. FGDs and key informant interviews were used to interview 70 women attending for routine ANC or PNC in the outpatient clinics at a large public hospital in Islamabad, Pakistan. The sampling frame included: (1) women who were pregnant or who have given birth within the past 6 weeks and (2) women who above the legal age of consent. Purposive sampling was used to approach women attending each study setting healthcare facility for routine ANC or PNC care. Women were only included once in the study. All interviews and FGDs were held in quiet locations away from the clinical areas at the relevant hospitals.

### Participants and settings
All women attending for routine ANC or PNC in the study healthcare facilities were invited to participate. The study settings were two large public urban hospitals serving women with low-lower middle socioeconomic status; and with varied education levels.[4] Similar proportions of women were included from four different stages of pregnancy and FGDs included women within the same assessment stage of pregnancy: first half of pregnancy (≤20 weeks); second half of pregnancy (>20 weeks); early postnatal stage (day 1–7) or late postnatal stage (week 2–6). Women were purposively selected and recruited sequentially until the number of anticipated FGD and key informant interviews was reached, and data saturation was achieved.

### Topic guide
A topic guide was developed to guide the interviews and was piloted in the UK with clinical researchers from LMIC (online supplementary document 1). The topic guide was then translated into the local study setting language, back translation was conducted, and the topic guide was further piloted in each study setting. As a result, the topic guide was refined; for example, some wording was changed in the local language and the introduction was amended to ensure that the women were aware that we sought to assess their general views and not their own personal experiences (if any) of different types of ill health. The topic guide served as a flexible tool to aid the facilitators in obtaining the women's opinions while ensuring that the interview remained on topic. The topic guide also acted as a cue to ask more probing questions to further understand women's beliefs and perceptions. The topic guide included three main subject areas:
1. Perception and understanding of health and ill health in general.
2. Perception and understanding of health and ill health during and after pregnancy.

3. Views on enquiry regarding mental and social ill health during and after pregnancy; and suggestions on how best to approach all components of health.

## Data collection

Prior to interview, all eligible women were approached and given verbal and written information regarding the study including a brief overview of the research aims and interview questions. Appointments for FGD or key informant interviews were then scheduled for a later time convenient for the women. The FGD and key informant interviews were conducted in the local language (India-Hindi and Pakistan-Urdu) by the local female trained and experienced researchers (ARA, RA and SS). On average, the FGDs lasted 60–90 minutes, and the key informant interviews lasted 30 minutes. All FGD and key informant interviews were conducted face to face, recorded on a digital recording device and transcribed in the local language on completion by the local researchers (ARA, RA and SS). All efforts were made to emphasise confidentiality for interviews to ensure women were confident in giving honest answers and member checking was used to assure correct interpretation. Interviewing women from different stages of pregnancy, different backgrounds and different countries broadened the scope of the topic and enabled transferability to be achieved to an extent.

## Data analysis

In each study setting, transcribed interviews were translated into English and initially manually open coded following data familiarisation (reading and rereading of the transcripts several times) by the local researchers (RA in India; ARA and SS in Pakistan). In a subsample of cases and prior to translation, participants were asked to review the coded transcripts to verify the accuracy and appropriateness of the applied code. Using thematic analysis, codes were identified from the transcripts and grouped into categories independently by the local researchers, enabling the first abstraction of data in each setting.[11] Thematic framework analysis of the categories was then undertaken. The separate results were then brought together and refined to agree on the key themes. All of the coded data was reviewed by other members of the research team (MM for India and SZ for Pakistan) to ensure consistency and to check for inter-rater reliability. In addition, all members of the research team reviewed, and critiqued emerging themes and subthemes and any discrepancies were discussed, and consensus agreed. The reporting of the results adheres to the Consolidated criteria for Reporting Qualitative research guidelines.[12]

## Patient and public involvement

No patient nor members of the public were involved in the design of this study.

## RESULTS

A total of 130 women were interviewed (table 1). Three women invited to participate declined due to time constraints. A total of 83 women were pregnant at the time of interview and 47 had given birth with the past 6 weeks. Most women were either in their first pregnancy or had one previous pregnancy. A minority of women had more than two previous pregnancies. The age range was from 20 to 36 years, with a mean of 26 years old. There were no distinct differences in emerging themes or subthemes from women from India or Pakistan. Therefore, all quotes illustrating the results are combined for both countries and displayed in tables and linked to the narrative text.

### What is health?

Women perceived 'good health' in general as: having a good diet; having an ideal weight; absence of physical disease; absence of stress and tension and a supportive family environment (table 2). Most women expressed that the ability to have a good diet would result in good health (table 2, Q1–Q3). Weight was mentioned in India and Pakistan, however, some women's perception of this differed (table 2, Q4–Q5). In both settings, women reported that health meant lack of disease and good energy levels (table 2, Q6–Q7). Most women reported that

**Table 1** Participants

| Stage of pregnancy | India | | Pakistan | | |
| | Method | Number of women | Method | Number of women | Total number of women |
| --- | --- | --- | --- | --- | --- |
| First half of pregnancy (≤20 weeks) | 2 FGD | 16 | 2 FGD | 26 | 42 |
| Second half of pregnancy (>20 weeks) | 3 FGD | 17 | 2 FGD | 24 | 41 |
| Early postnatal stage (day 1–7) | 3 FGD | 22 | 10 interviews | 10 | 32 |
| Late postnatal stage (week 2–6) | 1 FGD | 5 | 10 interviews | 10 | 15 |
| **Total** | | **60** | | **70** | **130** |

FGD, focus group discussions.

**Table 2** Women's understanding of health and ill health in general

| Health Subtheme | | Illustrative quotes |
|---|---|---|
| Diet | Q1 | 'If a person is happy but not eating well, how can she be healthy? Diet is so important.' (FGD 1, ANC, Pakistan) |
| | Q2 | 'Good health is good food.' (Interview 6, PNC, Pakistan) |
| | Q3 | 'Health means to have a balanced diet.' (FGD 5, ANC, India) |
| Weight | Q4 | 'One should be strong and a bit bulky.' (FGD 1, ANC, India) |
| | Q5 | 'Healthy women are those who are slim and have strength.' (Interview 3, PNC, Pakistan) |
| Energy | Q6 | 'Good health is when there is no disease.' (FGD 3, ANC, Pakistan) |
| | Q7 | 'Women should have the energy to do the chores, that means they are healthy.' (FGD 4, ANC, Pakistan) |
| Mental health | Q8 | 'Being healthy means having no stress or worries and having a tension-free life.' (FGD 5, ANC, India) |
| | Q9 | 'There is health of the mind and other body parts. If one body part is missing or is not healthy, one cannot be considered healthy. Similarly, if one is not mentally healthy, one is not able to do anything.' (FGD 1, ANC, Pakistan) |
| | Q10 | 'Good health means being mentally, socially, emotionally and physically fit.' (FGD 2, ANC, India) |
| Social health | Q11 | 'If there is peace in the home and in the mind, then it is good health.' (Interview 3, PNC, Pakistan) |
| | Q12 | 'If there are fights in the home, this effects a woman's health negatively.' (Interview 8, PNC, Pakistan) |
| | Q13 | 'The surrounding environment and family atmosphere directly affects [physical] health.' (FGD 4, PNC, India) |
| **Ill health Subthemes** | | **Illustrative quotes** |
| Physical symptoms | Q14 | 'Ill-health means presence of disease.' (FGD 4, ANC, India) |
| | Q15 | 'Those who are not healthy, their face shows it; they have yellowish complexion and have dark circles around the eyes.' (FGD 1, ANC, Pakistan) |
| | Q16 | 'Ill-health is like flu, fever and sore throats.' (Interview 5, PNC, Pakistan) |
| | Q17 | 'Poor health means having pain, discomfort, fever and body ache.' (Interview 4, PNC, Pakistan) |
| | Q18 | 'Disease is for example HIV/AIDS, weak bones, weakness.' (Interview 5, PNC, Pakistan) |
| Mental ill health | Q19 | 'Ill-health is about tension; even if there is a very low level of stress or tension it does great damage to a person.' (Interview 2, PNC, Pakistan) |
| | Q20 | 'Mental health affects physical health. If one is mentally disturbed, won't be able to do anything, even if you are physically fit, you will feel weak.' (Interview 9, PNC, Pakistan) |

ANC, antenatal care; FGD, focus group discussions; PNC, postnatal care.

health was not just physical health but includes mental health (table 2, Q8–Q10). Most women highlighted that a supportive environment with family were important for good health (table 2, Q11–Q13).

## What is ill health?
The emerging themes related to perceived ill health included: physical symptoms and disease; tension; and an abusive and stressful home environment. Many women expressed that ill health was 'pain, influenza, fever, sore throat, weakness, body ache, headache and swelling' (table 2, Q14–Q17). Ill health was also described in terms of disease, for example, HIV/AIDS (table 2, Q18).

Mental ill health was often considered a health concern and stress and tension was reported by many women as a component of, or factor leading to mental ill health (table 2, Q19–Q20).

## What is health during and after pregnancy?
During and after pregnancy a nutritious diet, medication, rest, extra care and a peaceful home environment were mentioned as important. Most women reported that diet is even more important when a woman is pregnant (table 3, Q21–Q22). Many women mentioned that pregnant women needed medication and more care from the family and healthcare providers (table 3, Q23–Q25).

**Table 3** Women's understanding of health and ill health during and after pregnancy

| Health Subtheme | | Illustrative quotes |
|---|---|---|
| Diet | Q21 | 'In pregnancy, dietary care is more important to remain healthy, as there is a baby as well.' (FGD 4, ANC, India) |
| | Q22 | 'One needs to be extra careful and eat more in pregnancy because it is the matter of two lives now.' (Interview 2, PNC, Pakistan) |
| Medication | Q23 | 'Good health in pregnancy is when you take your medicine regularly and completely.' (Interview 4, PNC, Pakistan) |
| Extra care is required during and after pregnancy for health | Q24 | 'You have to take really good care of yourself in pregnancy. If you become weak, the complications increase, and the baby will also be weak.' (Interview 4, PNC, Pakistan) |
| | Q25 | 'Pregnant woman must take a nap for one and a half hour to two hours during daytime and try to take more rest at night.' (FGD 2, PNC, India) |
| | Q26 | 'The most important thing is this that baby is healthy and to have the proper medical care so that both baby and mother stay healthy.' (Interview 7, PNC, Pakistan) |
| Mental components | Q27 | 'Women should not have any tension during pregnancy.' (FGD 3, ANC, India) |
| Social components | Q28 | 'If family members are quarrelsome and abusive, it affects both the mother and baby's health.' (FGD 2, ANC, India) |

| Ill health Subtheme | | Illustrative quotes |
|---|---|---|
| Physical symptoms | Q29 | 'Ill-health is …. abdominal pain, leaking per vagina, vomiting, loss of appetite and dizziness.' (FGD 2, ANC, India) |
| | Q30 | 'There are many symptoms that are dangerous, like if there is lack of fluids or baby around the baby is not growing as it should be, or if baby is suffering due to negligence.' (Interview 5, PNC, Pakistan) |
| | Q31 | 'Bleeding or swelling is dangerous.' (FGD 3, ANC, Pakistan) |
| Mental components | Q32 | 'Stress affects health in pregnancy, if there is any stress or tension one cannot eat properly. It also affects the baby's health and our own health too.' (Interview 4, PNC, Pakistan) |
| Social components | Q33 | 'If family members are quarrelsome and abusive, it affects both the mother and baby's health.' (Interview 7, PNC, Pakistan) |
| | Q34 | 'The husband comes home after drinking and beats the woman.' (FGD 2, ANC, India) |
| | Q35 | 'If a husband spends his entire income on drinking, he comes home late and beats his pregnant wife.' (FGD 4, ANC, India) |
| Gender preference of the baby | Q36 | 'In some families, male gender preference is a major cause of stress and tension for pregnant women.' (FGD 1, ANC, India) |
| | Q37 | 'This time also I have given birth to a girl child [and] because of this the family atmosphere is not good. My family members are abusing me and because of this I am going through a lot of stress.' (FGD 5, PNC, India) |

ANC, antenatal care; FGD, focus group discussions; PNC, postnatal care.

Many women understood that their health was 'lack of tension', and that their health (or ill health) was linked to that of their baby (table 3, Q26–Q28).

### What is ill health during and after pregnancy?

Perceived ill health during and after pregnancy included the perception that pregnancy is a 'disease', and that there was 'no health in pregnancy', as all 'the energy goes to the baby and the mother becomes weaker'. Women also considered some physical symptoms and discomfort as 'normal' during the pregnancy. This included nausea and vomiting, back pain and difficulty sleeping. However, many women reported that ill health during and after pregnancy was 'vomiting, high blood pressure, bleeding, swelling, abdominal pain' (table 3, Q29). Many women were aware that these symptoms could be dangerous (table 3, Q30–Q31). Stress and tension were considered important components of ill health by most women (table 3, Q32). An abusive family environment was highlighted as examples of social ill health in women during and after pregnancy (table 3, Q33–Q35). There was some discussion regarding gender preference, and it was suggested by a minority of women that the gender of a baby created stress and tension for some women (table 3, Q36–Q37).

### Should healthcare providers ask about mental and social ill health?

An emerging theme regarding what women consider health and ill health to be, was the importance of mental and social health and well-being. Women described mental ill health such as 'depression', 'weeping', 'stress', 'tension', 'grief', 'loss of appetite' and 'unhappiness'. Women reported social concerns such as 'fights in the home', 'domestic violence', 'poor family environment', 'quarrels and verbal abuse', 'family or husband alcohol misuse' and an 'expectation of a male child'. Women were, therefore, asked questions regarding how best to approach and address the concepts of mental and social ill health for women during and after pregnancy. Most

women reported that healthcare providers currently ask pregnant women questions regarding physical ill health only (table 4, Q38–Q40). Most women felt that there was a lack of enquiry from healthcare providers regarding mental and social ill health during and after pregnancy. Most women felt that healthcare providers could and should ask women about their mental and social health (table 4, Q41–Q43). Many women said that it would be beneficial to speak to their healthcare provider if they were feeling 'stressed' and 'under tension' as the healthcare provider might be able to provide general support and advice. However, a few women explained these types of questions might be considered unacceptable, because of lack of education and understanding among some

**Table 4**  Women's views regarding whether healthcare providers should enquire regarding mental and social ill health

| Subtheme | | Illustrative quotes |
|---|---|---|
| Mental and social ill health is not currently addressed | Q38 | 'They do not ask about mental health or the family situation. They just ask about test reports, blood deficiency' (FGD 4, ANC, Pakistan) |
| | Q39 | 'They ask in detail about me, about my diet and medicines, they take a complete history and ask about all my previous deliveries, but they never ask about my family environment or mental health.' (Interview 10, PNC, Pakistan) |
| | Q40 | 'They usually don't ask much, and we also do not talk much; they check, give medicine, or sometimes ask, does your husband smoke and is there any disease in the family?' (Interview 7, PNC, Pakistan) |
| Mental and social ill health should be assessed | Q41 | 'Yes, they should ask us.' (FGD 4, ANC, India) |
| | Q42 | 'They should ask such questions, but they do not. They should ask about family conflicts and talk about the solutions.' (Interview 11, PNC, Pakistan). |
| | Q43 | 'If one has some grief, sharing with someone lessens the grief, though it may not solve the problem, but releases stress.' (Interview 6, PNC, Pakistan) |
| Education | Q44 | 'Women who are uneducated or unaware might get offended by these questions, others will not mind.' (FGD 2, ANC, Pakistan) |
| | Q45 | 'People who understand such issues will not react, but others will.' (FGD 3, ANC, India) |
| Stigma | Q46 | 'Some women will not talk about their problems as they are afraid. They think that their husband will know about it if they talk to doctor and will be angry with them.' (FGD 4, ANC, Pakistan) |
| | Q47 | 'I will feel ashamed to talk about my condition... I can tolerate until it is really necessary.' (Interview 8, PNC, Pakistan) |
| | Q48 | 'Some women don't share. Some women keep everything inside.' (Interview 11, PNC, Pakistan) |
| Should not be assessed | Q49 | 'Doctors should not ask these questions, no, they should not.' (Interview 5, PNC, Pakistan) |
| | Q50 | 'No, doctor has no right to ask such questions.' (Interview 6, PNC, Pakistan) |
| | Q51 | 'A doctor has many patients ...Obviously, they don't have time for these details!' (FGD 4, ANC, Pakistan) |
| Impact on family | Q52 | 'If you ask about the family situation, our community is such that women will be afraid as to why these questions are being asked and they have a fear that in-laws will find out' (FGD 2, ANC, Pakistan) |
| | Q53 | 'If the doctor asks in front of our family, that will help change bad behaviours in the family' (FGD 3, ANC, Pakistan) |
| Solutions | Q54 | 'They should use questions in a way that are focused and clear.' (FGD 3, ANC, Pakistan) |
| | Q55 | 'Do you have any tension? How is your home environment? How is your husband and your family-in-laws behaviour with you?' (Interview 10, PNC, Pakistan) |

ANC, antenatal care; FGD, focus group discussions; PNC, postnatal care.

women (table 4, Q44–Q45). There was some concern that some women would prefer not to talk about their mental and social ill health as they may feel afraid, ashamed or uncomfortable (table 4, Q46–Q48). A small minority of women in Pakistan felt it was not the role of the healthcare provider to ask questions regarding mental and/ or social health, as this was their 'personal life'. Women discussed the repercussions from husbands and family if domestic violence during and after pregnancy would be disclosed by a woman to a healthcare provider (table 4, Q49–Q51). There were concerns that a woman would not be allowed to access care again. There was a lot of discussion about this with sometimes conflicting opinions; ranging from it not being right for a woman to disclose domestic violence, or mental ill health such as stress and tension; to a consideration that healthcare providers should ask and would provide help. A few women said that if they themselves were asked questions regarding depression or domestic violence, they would seek care from another healthcare provider to avoid these questions. Many women recognised that there were existing barriers for healthcare providers which made it difficult for them to ask women questions regarding mental and social ill health during and after pregnancy (table 4, Q52–Q53). Such barriers included the perceived heavy workload of healthcare providers and lack of time to explore mental and/or social concerns (table 4, Q51). Other women explained that 'they can ask, but what can they do?' and that 'nothing can be done' to help. Some women did offer suggestions of how healthcare providers could best ask questions regarding mental and social ill health during and after pregnancy. A more empathic approach to women was suggested including politeness and friendliness. Confidentiality was highlighted as important by many women. Some women suggested that healthcare providers need not go into such detailed questions but keep any questions simple (table 4, Q54–Q55).

## DISCUSSION
### Statement of principal findings
Most women who took part in this study had clear opinions and had a good understanding that health includes more than physical components, and that the health of the mother was linked to that of the baby. Good diet, healthy weight, rest and a supportive home environment contributed to good health; whereas physical disease symptoms, complications of pregnancy, as well as stress, tension and an abusive family environment contributed to ill health. Women were aware of physical common discomforts and of more severe complications and dangers signs, as ill health related to pregnancy. Women considered mental and social ill health as important but reported that healthcare providers do not currently ask women how they are feeling or ask regarding their mental nor social health and well-being . However, most women would be happy if healthcare providers did so. A minority of women considered mental and social health concerns

as 'personal' and reported that there may be barriers to asking such questions.

### Strengths of the study
To the best of our knowledge, this is the first study from two LMIC settings (India and Pakistan) to obtain women's own opinions on what they might consider health and ill health to be in general and during and after pregnancy. A range of women from four different stages of pregnancy were interviewed resulting in a wide spectrum of responses. Most women approached welcomed the discussion surrounding health, and ill health during and after pregnancy, and were keen to contribute to solutions in their settings regarding healthcare providers roles in routine enquiry regarding other components of health including mental and social aspects, and not just physical health.

### Limitations of the study
This study population included women attending for routine ANC or PNC in public secondary level healthcare facilities in urban settings and excludes other women who may have alternative perspectives or different insights. There is a need to assess the views of women in other LMIC and including community and rural settings, as these women may have different perceptions and understandings.

### How does this study relate to other literature?
There is lack of literature regarding the understanding of what women consider health and ill health to be during and after pregnancy in India and Pakistan. There is also debate that the current definition of health[2] is outdated and needs to be reformulated to consider health in a context of functionality, capacity, adaptability and the ability to perform activities of daily living despite having a disease or disability.[13] There have been various proposals to develop the definition of health using the concept of 'health, as the ability to adapt and to self-manage' but with a continued emphasis on the importance of the three domains of health: physical, mental and social.[13] Aspects of physical ill health in women during and after pregnancy in LMIC settings are well documented and there are many recommendations and interventions developed to diagnose and manage these conditions; for example, anaemia, haemorrhage, hypertensive disorders of pregnancy, infection and complications of obstructed labour.[14–18] With regard to mental and social ill health in women during and after pregnancy in LMIC, there is emerging data to demonstrate that depression and domestic violence represent a large burden of ill health and have a negative impact on a woman's health and well-being.[19–22] However, to date, clinical practice in low-resource settings rarely takes mental or social ill health into account when assessing a woman's health needs during and after pregnancy.[23 24] However, this study demonstrates that most women would welcome such an assessment but report that their healthcare providers do

not current ask; and even if they did, some women report that 'nothing that can be done'. This may be due to a shortage of human resources but also because of healthcare providers' perceptions that these issues are not 'ill health' per se and because of perceived cultural sensitivities related to mental ill health and social concerns.[4 23–26]

It is well recognised that maternal physical, mental and social ill health are linked to poor fetal and newborn health outcomes and women in our study demonstrated an understanding of this important association.[27–30] The detrimental impact of mental and social morbidity on the overall health and well-being of mothers (and their babies) during and after pregnancy has resulted in public health policy in high-income countries such as the UK, where enquiry regarding of mental ill health (depression, anxiety) and social circumstances (exposure to domestic violence and/or substance misuse), is routinely conducted during and after pregnancy.[31] WHO has produced clinical and policy guidelines on how to comprehensively care for women during and after pregnancy including identification, safety assessment and planning, communication and clinical skills, documentation and provision of referral pathways for mental and social ill health.[32] However, the feasibility of implementation and acceptability of this guidance in countries, such as India and Pakistan, are currently uncertain. There is debate as to the cadre of healthcare providers most suitable to undertake routine enquiry regarding for and/or manage mental and social ill health in LMIC.[23 24 33] In many high-income countries, specially trained midwives routinely assess, screen, support and provide further referral for women with mental and social morbidity.[31] This study shows that most women are keen to be asked questions regarding their mental and social well-being during and after pregnancy, as part of routine maternity care. However, a minority of women do not wish to be asked.

Currently, there is good coverage and uptake of ANC, but content needs to be adapted to ensure good quality comprehensive care that is respectful, integrated and delivers physical, mental and social care, with agreed country specific additional content. Although, mental and social ill health is clearly a real matter of concern for women, this study was not designed to explore this in further detail. Further research is needed to explain in more depth what these problems are, how they can be addressed, and by whom. It is important that healthcare providers are supported to provide good quality care for women and that this goes beyond simply the physical aspects of health and is inclusive of mental and social well-being.

## CONCLUSION

Women reported that health and ill health is not just physical but also includes mental and social components. Maternal ill health is common among women attending for ANC and PNC.[4] Women are increasingly accessing care during and after pregnancy and there is a window of opportunity to adapt and amend available care packages to include comprehensive enquiry regarding and, where needed, support for mental and social ill health.[4 10] This study demonstrates that most women are keen to be asked questions regarding their mental and social well-being and provides simple suggestions that healthcare providers can use to improve the enquiry regarding mental and social ill health among women during and after pregnancy. There is a need for further research to help develop interventions and deliver maternity care in a way that meets the health needs of women in a comprehensive and holistic manner, including mental and social components.[4 5]

**Acknowledgements** The authors would like to thank Pratima Mittal, Jyotsna Suri and Mansi for their support with ethics applications and data collection in India. A huge thank you to all the women who participated in this study.

**Contributors** MM conceived the study idea, design and developed the topic guide. SZ coordinated and supervised the research activities in Pakistan. ARA, SS and RA conducted the interviews, transcription, translations and data analysis. MM coordinated and supervised the overall research activities, interpreted and presented the results and wrote the manuscript. NvdB has reviewed the data, interpreted the results and contributed to the manuscript. All authors have read, edited and approved the final manuscript for submission.

**Funding** Disclosure of funding received: This study was funded by the Liverpool School of Tropical Medicine Research Development Fund (RDF160301MMC11).

**Competing interests** None declared.

**Patient and public involvement statement** No patient nor members of the public were involved in the design of this study.

**Patient consent for publication** Not required.

**Ethics approval** The Liverpool School of Tropical Medicine, Liverpool, UK, granted full ethical approval (LSTM 16-054RS). Ethical approval was also obtained from the Research and Ethics Committee, Vardhman Mahavir Medical College and Safdarjung Hospital, New Delhi, India (ICE/SJH/VMMC/Project/October-2016/688); and the National Bioethics Committee, Islamabad, Pakistan (No.4-87/16/NBC-159-Addendum/RDC/3634).

**Provenance and peer review** Not commissioned; externally peer reviewed.

**Data availability statement** Data are available on reasonable request.

**ORCID iD**
Mary McCauley http://orcid.org/0000-0003-1446-0625

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
