## [Reviewer comments · BMJ Open]

ARTICLE DETAILS

TITLE (PROVISIONAL)	“Good health means being mentally, socially, emotionally and physically fit”: Women’s understanding of health and ill-health during and after pregnancy in India and Pakistan: a qualitative study
AUTHORS	McCauley, Mary; Avais, Ayesha; Agrawal, Ritu; Saleem, Shumaila; Zafar, S; van den Broek, Nynke

VERSION 1 – REVIEW

REVIEWER	Caroline Homer Burnet Institute, Australia
REVIEW RETURNED	28-Jan-2019

GENERAL COMMENTS	Much of the results and discussion seem to focus on mental health screening. This is not really mentioned in the abstract or article summary. It would be useful to include this more explicitly. I do not find the acronym KII helpful as I had to keep trying to remember what it meant. I suggest writing this out fully as it is not a commonly used abbreviation. The data from 2 countries are combined It is likely that these contexts have differences and so the validity of combining them needs to be discussed in the limitations.
--

REVIEWER	Nancy Byatt, DO, MS, MBA, FACLP University of Massachusetts Medical School Worcester, MA USA
REVIEW RETURNED	09-Apr-2019

GENERAL COMMENTS	This manuscript discusses women’s perspectives on maternal health in the perinatal period in India and Pakistan and their general experience with and attitudes towards receiving healthcare that includes “psychosocial” care. This article addresses very important and timely topics, namely maternal morbidity and mortality and has some very interesting and important findings. The use of thematic analysis, coupled with many powerful quotes from participants, makes for a striking read. Overall, I think this is an important contribution to the literature. Overall, I like the message of the introduction. However, it could use some tightening as well as conceptual restructuring. Mainly, I think the authors spend too much time discussing maternal health generally, but very little on the general knowledge (or lack thereof) and policies of maternal health in India and Pakistan and other LMIC. I also don’t think the authors need to expend 3 paragraphs
--

	convincing readers of the risks of maternal ill-health generally – I think this can be made more succinctly and that readers will generally be in agreement! Spending more time on the views of ill-health in low- and middle-income countries would be more useful, since this is minimally discussed in the intro and is the main subject of the article. Also, after reading the results, it would be helpful to have a better understanding of women’s general levels of education and health literacy in these areas, to provide better context for their responses. The methods section needs more discussion on recruitment. What were the inclusion/exclusion criteria? Please describe more about the study setting including the locations of these facilities from which women were recruited and the population served. Was an IRB used and, if so, how was informed consent obtained? Additionally, in the data analysis section, more clarity should be given as to the procedures and analysis. For example, they say that transcripts were reviewed by a second reader for “sense checking and to avoid bias.” How specifically was this done? Were there any inter-rater reliability scores that can be reported for the independent coding done in the thematic analysis? What software program was used for the analysis (e.g., NVivo, Dedoose)? The results are well put together. If available, inclusion of percentages or raw numbers for some statements would be helpful, since there is no corresponding table (e.g., XX% of women had more than 2 previous pregnancies; XX% of women expressed that the ability to have a good diet would result in good health.) My main qualm with the results is the length: though interesting and very important, some of the messages are lost in the large amount of text. It seems like some of the quotes could be omitted, or even put in an appendix with a more expansive list of quotes. For example, I would remove those that are not as striking or that so closely mirror the thematic element, that repeating as a quote seems almost redundant. For example, line 15, page 9 – “good health is from a good diet” is almost exactly the description of this group. Some of these can be removed without losing the gist of the argument. I found the discussion section to be the weakest section of the article. The large section on how this study relates to the literature could be amended to weave in more of the important findings from this study. The finding that many women were not asked about their psychological health is very important. Additionally, it is exciting and important that women feel that this is a necessary part of their care and that providers should be doing this! Another important finding is the perception that even if they did bring it psychosocial or mental health concerns, nothing could be done about it. In my opinion, the most novel findings here are that: 1) women are yearning for more holistic care which addresses their psychosocial situation and mental health and, 2) women perceive that their providers are not asking about that and that is even if did, there is nothing that can be done. These finding suggest the need to for building the capacity of front-line providers to: 1) perform psychosocial and mental health assessments and, 2) have a plan in place for how to respond. Please discuss this in the discussion. Please spend more time discussing the implications of this and other results, rather than just a line or two. I would also encourage the authors to comment on how their findings can inform future studies or program focused on improving maternal and child health (e.g.,
--	---

	integrating psychosocial mental health assessments and treatments into maternity care). I think this would greatly strengthen the quality and impact of the article. Similarly, please revise the summary and abstract with this in mind. More minor  - In the introduction, the introduction of SDG needs more context as well, as not all readers will be familiar with this initiative. - Line 20, page 6: ANC & PNC need to be defined before using acronyms - Line 24, page 10: the use of semicolons is confusing here – would use commas - Line 41/42, page 13: I believe there a typo (“healthcare providers did not enquire...)
--	---

VERSION 1 – AUTHOR RESPONSE

Reviewer 1

Much of the results and discussion seem to focus on mental health screening. This is not really mentioned in the abstract or article summary. It would be useful to include this more explicitly. Thank you. The need for mental health screening emerged as one of the main themes from the interviews. The abstract and article summary has been amended to further highlight mental health screening.

I do not find the acronym KII helpful as I had to keep trying to remember what it meant. I suggest writing this out fully as it is not a commonly used abbreviation. Thank you. This acronym has been deleted in the text of the document and spelt out in full in the text. However, I have included this acronym for the tables and quotes for ease of readability.

The data from 2 countries are combined It is likely that these contexts have differences and so the validity of combining them needs to be discussed in the limitations. Thank you. This point has been explained further in the results section.

Reviewer 2

This manuscript discusses women’s perspectives on maternal health in the perinatal period in India and Pakistan and their general experience with and attitudes towards receiving healthcare that includes “psychosocial” care. This article addresses very important and timely topics, namely maternal morbidity and mortality and has some very interesting and important findings. The use of thematic analysis, coupled with many powerful quotes from participants, makes for a striking read. Overall, I think this is an important contribution to the literature. Thank you.

Overall, I like the message of the introduction. However, it could use some tightening as well as conceptual restructuring. Mainly, I think the authors spend too much time discussing maternal health generally, but very little on the general knowledge (or lack thereof) and policies of maternal health in India and Pakistan and other LMIC. I also don’t think the authors need to expend 3 paragraphs convincing readers of the risks of maternal ill-health generally – I think this can be made more succinctly and that readers will generally agree! Spending more time on the views of ill-health in low- and middle-income countries would be more useful, since this is minimally discussed in the intro and is the main subject of the article. Also, after reading the results, it would be helpful to have a better

understanding of women's general levels of education and health literacy in these areas, to provide better context for their responses.

Thank you. To date there is a lack of understanding and lack of data regarding what women themselves consider ill-health to be (physical and/or psychological and/or social) and their experiences (subjective measures) compared to clinical findings (objective measures) when assessed and defined as "morbidity" by a healthcare provider. The World Health Organization have recently highlighted the need for qualitative studies that explore women's lived experiences of ill-health and perceptions of their health and health needs during and after pregnancy in low-and middle-income countries. The introduction has been amended to include some of the suggested improvements.

The methods section needs more discussion on recruitment. What were the inclusion/exclusion criteria?

Thank you. The inclusion/exclusion criteria has been included in the text of the methodology.

Please describe more about the study setting including the locations of these facilities from which women were recruited and the population served.

Thank you. The study settings have been included in the text of the methodology.

Was an IRB used and, if so, how was informed consent obtained?

Yes, full ethical approval was obtained in Liverpool UK, in Islamabad Pakistan and New Delhi, India. The text has been amended to include this detail and that written consent was obtained from each woman involved in the study.

Additionally, in the data analysis section, more clarity should be given as to the procedures and analysis. For example, they say that transcripts were reviewed by a second reader for "sense checking and to avoid bias." How specifically was this done? Were there any inter-rater reliability scores that can be reported for the independent coding done in the thematic analysis? What software program was used for the analysis (e.g., NVivo, Dedoose)?

Thank you. The text has been amended to ensure the data analysis process is clear.

The results are well put together. If available, inclusion of percentages or raw numbers for some statements would be helpful, since there is no corresponding table (e.g., XX% of women had more than 2 previous pregnancies; XX% of women expressed that the ability to have a good diet would result in good health.)

Thank you. Using percentages or raw numbers is not part of qualitative analysis.

My main qualm with the results is the length: though interesting and very important, some of the messages are lost in the large amount of text. It seems like some of the quotes could be omitted, or even put in an appendix with a more expansive list of quotes. For example, I would remove those that are not as striking or that so closely mirror the thematic element, that repeating as a quote seems almost redundant. For example, line 15, page 9 – "good health is from a good diet" is almost exactly the description of this group. Some of these can be removed without losing the gist of the argument. Thank you for this suggestion. I have edited the results section and put quotes into tables to ensure that the key messages are not lost in the text. This has helped decrease the word count also.

I found the discussion section to be the weakest section of the article. The large section on how this study relates to the literature could be amended to weave in more of the important findings from this study. The finding that many women were not asked about their psychological health is very important. Additionally, it is exciting and important that women feel that this is a necessary part of their care and that providers should be doing this!

Thank you. We have amended the discussion section to highlight that women are not currently asked regarding their psychological health but would be keen for this happen.

Another important finding is the perception that even if they did bring it psychosocial or mental health concerns, nothing could be done about it. In my opinion, the most novel findings here are that: 1) women are yearning for more holistic care which addresses their psychosocial situation and mental health and, 2) women perceive that their providers are not asking about that and that is even if did, there is nothing that can be done. These finding suggest the need to for building the capacity of front-line providers to: 1) perform psychosocial and mental health assessments and, 2) have a plan in place for how to respond.

Please discuss this in the discussion. Please spend more time discussing the implications of this and other results, rather than just a line or two.

Thank you. We have amended the discussion section to highlight these points.

I would also encourage the authors to comment on how their findings can inform future studies or program focused on improving maternal and child health (e.g., integrating psychosocial mental health assessments and treatments into maternity care). I think this would greatly strengthen the quality and impact of the article.

Thank you. As we mention in the discussion, there is lack of evidence as to what works best for psycho-social screening in low resource settings. There is an ongoing ethical debate whether screening should be routine conducted in low resource settings, as women identified with psycho-social needs may not have access to treatment, support, management or referral for such problems due to lack of health service infrastructure. We highlight the need for further implementation research into these areas and the text has been amended to include this point.

Similarly, please revise the summary and abstract with this in mind.

Thank you. The text has been amended to include all of the above points.

More minor

- In the introduction, the introduction of SDG needs more context as well, as not all readers will be familiar with this initiative.

Thank you. Text has been amended.

- Line 20, page 6: ANC & PNC need to be defined before using acronyms

Thank you. Text has been amended.

- Line 24, page 10: the use of semicolons is confusing here – would use commas

Thank you. Text has been amended.

- Line 41/42, page 13: I believe there a typo (“healthcare providers did not enquire...)

Thank you. Text has been amended.

VERSION 2 – REVIEW

REVIEWER	Nancy Byatt, DO, MS, MBA UMass Medical School USA Dr. Byatt's disclosures are attached below as a word document.
REVIEW RETURNED	14-Jun-2019

GENERAL COMMENTS	Thank you for the opportunity to review this article a second time. It was exciting to see the improvements made by the authors and I found the manuscript to be much better the second time around. I appreciated the shortening of the introduction and making the message more cogent. I do still think it could be made even more specific to that of LMIC and therefore the importance of this work, given that they are underrepresented.
---

	The methods section is vastly improved with the inclusions of more information around setting and participants, ethical procedures, and data analyses. It was also very helpful to have the supplement and see the structure of the interview itself. Of note, “setting” is used in 2 consecutive sub-headings. If there are any kappa statistics or quantitative measures of rater reliability, it would be great to see these in there as well. I also appreciated how the results have been restructured and that the quotes have been moved into their own tables. This makes it much easier to read and does not seem like an inundation of data. The discussion section is improved as well and has been made more persuasive. I appreciate that it is now more emphasized that many women are open to care and thus this is the duty of healthcare workers and there is a need for further research in this area. I think that the strengths and limitations sections might be moved to right before the conclusions, though, so that some of this is emphasized earlier. Additionally, many of the paragraphs throughout but especially in the discussion are quite long; the authors may consider breaking up into multiple where available. Finally, I would suggest updating the abstract to be more aligned with your new discussion, including emphasizing that providers should be able and expected to address women’s mental health needs, especially as they are open to it, and we need to focus efforts on helping them to do so.
--	--

VERSION 2 – AUTHOR RESPONSE

Reviewer: 2	Response to reviewer’s request:
Thank you for the opportunity to review this article a second time. It was exciting to see the improvements made by the authors and I found the manuscript to be much better the second time around. I appreciated the shortening of the introduction and making the message more cogent. I do still think it could be made even more specific to that of LMIC and therefore the importance of this work, given that they are underrepresented.	Thank you. The text has been amended to make it even more specific to that of LMIC and therefore the importance of this work, given that they are underrepresented.
The methods section is vastly improved with the inclusions of more information around setting and participants, ethical procedures, and data analyses. It was also very helpful to have the supplement and see the structure of the interview itself.	Thank you.
Of note, “setting” is used in 2 consecutive sub-headings.	My sincere apologies. I have amended this text.
If there are any kappa statistics or quantitative measures of rater reliability, it would be great to see these in there as well.	Thank you. Unfortunately, we do not have any kappa statistics or quantitative measures of inter-rater reliability. I have however amended the text to comment on inter-rater reliability.
I also appreciated how the results have been restructured and that the quotes have been moved into their own tables. This makes it much easier to read and does not seem like an inundation of data.	Thank you.

The discussion section is improved as well and has been made more persuasive. I appreciate that it is now more emphasized that many women are open to care and thus this is the duty of healthcare workers and there is a need for further research in this area.	Thank you.
I think that the strengths and limitations sections might be moved to right before the conclusions, though, so that some of this is emphasized earlier. Additionally, many of the paragraphs throughout but especially in the discussion are quite long; the authors may consider breaking up into multiple where available.	Thank you. We are reluctant to move the strengths and limitations sections to right before the conclusions as we have followed the recommended structure of as per the British Medical Journal:  • Statement of principal findings • Strengths and weaknesses of the study • Strengths and weaknesses in relation to other studies, discussing particularly any differences in results • Meaning of the study: possible mechanisms and implications for clinicians or policymakers • Unanswered questions and future research
Finally, I would suggest updating the abstract to be more aligned with your new discussion, including emphasizing that providers should be able and expected to address women's mental health needs, especially as they are open to it, and we need to focus efforts on helping them to do so.	Thank you. I have updated the abstract to be more aligned with the discussion, including your suggestions.